# Health outcomes in Deaf signing populations: A systematic review

Katherine D. Rogers[1]*, Aleix Rowlandson[2], James Harkness[3], Gemma Shields[2], Alys Young[1]

1 Division of Nursing, Midwifery and Social Work, University of Manchester, Manchester, United Kingdom, 2 Division of Population Health, Health Services Research & Primary Care, University of Manchester, Manchester, United Kingdom, 3 Imperial College Healthcare NHS Trust, London, United Kingdom

* katherine.rogers@manchester.ac.uk

**Data Availability Statement:** All data are in the manuscript and/or Supporting information files.

**Funding:** This review is partly funded by Dr Katherine Rogers's NIHR (National Institute of Health and Care Research) Post-Doctoral

## Abstract

### Objectives

(i) To identify peer reviewed publications reporting the mental and/or physical health outcomes of Deaf adults who are sign language users and to synthesise evidence; (ii) If data available, to analyse how the health of the adult Deaf population compares to that of the general population; (iii) to evaluate the quality of evidence in the identified publications; (iv) to identify limitations of the current evidence base and suggest directions for future research.

### Design

Systematic review.

### Data sources

Medline, Embase, PsychINFO, and Web of Science.

### Eligibility criteria for selecting studies

The inclusion criteria were Deaf adult populations who used a signed language, all study types, including methods-focused papers which also contain results in relation to health outcomes of Deaf signing populations. Full-text articles, published in peer-review journals were searched up to 13th June 2023, published in English or a signed language such as ASL (American Sign Language).

### Data extraction

Supported by the Rayyan systematic review software, two authors independently reviewed identified publications at each screening stage (primary and secondary). A third reviewer was consulted to settle any disagreements. Comprehensive data extraction included research design, study sample, methodology, findings, and a quality assessment.

Fellowship (NIHR award reference number: PDF-2018-ST2-004). The views expressed in this publication are those of the author(s) and not necessarily those of the NIHR, NHS or the UK Department of Health and Social Care. There was no additional external funding received for this study.

**Competing interests:** The authors have declared that no competing interests exist.

## Results

Of the 35 included studies, the majority (25 out of 35) concerned mental health outcomes. The findings from this review highlighted the inequalities in health and mental health outcomes for Deaf signing populations in comparison with the general population, gaps in the range of conditions studied in relation to Deaf people, and the poor quality of available data.

## Conclusions

Population sample definition and consistency of standards of reporting of health outcomes for Deaf people who use sign language should be improved. Further research on health outcomes not previously reported is needed to gain better understanding of Deaf people's state of health.

## Introduction

This systematic review concerns the health outcomes of Deaf people who are sign language users. It identifies and evaluates available information in order to establish the state of our current knowledge as well as future research directions. Globally, WHO estimate that 466 million individuals are living with what WHO define as a "disabling hearing loss" with this figure expected to reach over 700 million by 2050 [1]. Of those it is estimated that over 70 million people use one of over 300 sign languages worldwide [2, 3]. Signed languages are not visual versions of the spoken language of a country or nation, they are separate, fully grammatical living languages in their own right [4]. Deaf individuals do not perceive being deaf as a disability, and together form a community, with their own distinct language, culture and history [5]. Conventionally the upper case 'Deaf' is used to distinguish them from the greater population of deaf people who are spoken language users and not affiliated with Deaf communities. This distinction between deaf and Deaf is not based on degrees of deafness in an audiological sense, but is rather a sociological distinction, based on cultural-linguistic identity.

Poorer health and mental health outcomes among Deaf communities have been previously observed when compared with the general population [6–9]. Suboptimal management of physical health conditions is also common, posing not just immediate health risk but increasing the risk of long-term complications. A UK study using the EQ-5D-5L recorded a mean health-state value of 0.78 for Deaf people compared to the mean health-state value for the general population of a similar age of 0.84 [10]. Common mental health problems have been found to be more prevalent amongst Deaf people in comparison with the hearing population [6]. Additionally, Deaf people are more likely to be victims of physical, sexual, and emotional abuse along with neglect, all of which are significant risk factors for poor mental health [11, 12]. Wide-spread difficulties in accessing health-related information in a signed language, accessing health care in a timely manner, cultural-linguistic barriers in interactions with clinicians and health care providers, and inappropriate diagnostic assessments normed on hearing populations have all been recorded as potential contributors to poorer health outcomes in this population [7, 13, 14]. Poor communication or no access to communication with healthcare professionals in a preferred language can leave Deaf patients with confusion surrounding their condition, or appropriate management techniques, thereby increasing their risk of poor or adverse health outcomes [8]. This can lead to late diagnoses, risky patient behaviours resulting from a lack of understanding and information, not having a

choice in treatment options, a lack of understanding of health conditions, with adverse consequences for patients.

In summary, the signing Deaf population experience inequalities in health outcomes, and in accessing and navigating healthcare. These inequalities are both widespread and multifactorial, and left unaddressed could be detrimental to the health of the signing Deaf population. A greater awareness and understanding of these issues across the healthcare community is paramount to improving service provision. This, combined with the fact that a comprehensive systematic review of the evidence concerning the physical and mental health of the signing Deaf adult population has yet to be undertaken, highlights the necessity for this review.

There are no similar reviews with a focus on the specificity of health outcomes. PROSPERO records a current systematic review of Inequities Experienced by Deaf and Hard of Hearing Patients in Healthcare Access and Healthcare Delivery [CRD42020161691] and one concerning the prevalence and correlates of mental and neurodevelopmental symptoms and disorders among deaf children and adolescents [CRD42020189403]. Neither addresses health and mental health outcomes of Deaf signing adults which is required as a guide to future research and to assist clinicians in their current work.

## Research questions

- What does the available literature conclude about the mental and physical health of adult Deaf population(s)?

  - How does the health of the Deaf population(s) compare to that of the general population (s)?

  - What are the strengths and weaknesses of the available literature?

  - What should future research aim to address?

## Methods

### Original protocol for the study

Prospero registration (S1 Appendix): https://www.crd.york.ac.uk/prospero/display_record.php?RecordID=182609.

### Search strategy

An electronic literature search was used to identify relevant studies up to (13th June 2023). No starting cut-off date was applied. The following electronic databases were searched using the OVID platform: Medline; Embase; PsychINFO; and Web of Science. The research strategy included the keywords (e.g. deaf*, health*, and sign*) and key terms were truncated and combined through use of the Boolean operators 'AND' and 'OR' (see S1 Table for full details).

### Study selection

Articles identified by the search underwent a two-step screening process: (i) Title and abstract screening against the inclusion/exclusion criteria in Table 1 below and (ii) Full text screening. Each stage of the screening was completed independently by two reviewers ([AUTHOR ONE] & [AUTHOR TWO]), with a third reviewer ([AUTHOR FIVE]) consulted to settle any disagreements. In the full-text screening, a list of reasons for exclusion were additionally recorded.

**Table 1. Inclusion and exclusion criteria.**

| Aspect of study | Inclusion | Exclusion |
|---|---|---|
| Population | • Signing Deaf populations*<br>• Adults (aged 18+)<br>• Studies focusing on a subgroup of Deaf populations (e.g. LGBTQ, those with learning disabilities/difficulties) | • deaf populations (with a lowercase 'd') and/or where deaf is not used to define those who are sign language users<br>• Hard-of-hearing populations who are not sign language users<br>• Children or adolescents<br>• Those with single sided deafness<br>• Individuals who are blind/ those with dual sensory impairment<br>• Individuals with age-related hearing loss<br>• Cochlear implant users who are not sign language users<br>• Deaf people who use spoken language exclusively |
| Study type | • All study types other than those in the exclusion criteria<br>• Full text<br>• Peer-reviewed<br>• Primary or secondary peer reviewed research articles | • Letters<br>• Editorial/opinion pieces<br>• Historical articles<br>• Case reports<br>• Conference abstracts |
| Outcomes | • Measures of mental health such as the prevalence of mental health conditions and measures used in relation to mental health (e.g., the PHQ-9 and GAD-7)<br>• Measures of physical health, such as the prevalence of chronic conditions and symptom measures for physical health conditions<br>• Measures of overall health status, quality of life and wellbeing, that may reflect mental and physical health combined<br>• Morbidity, mortality and prevalence statistics | • Papers reporting on the physical or mental health of carers/relatives of Deaf individuals<br>• Papers reporting on hearing health conditions including their aetiology and treatment<br>• Papers reporting studies on audiology<br>• Papers reporting on correction or improvement to hearing<br>• Papers reporting on health measurement instruments (such as questionnaire validation studies) without data on health status.<br>• Papers reporting on barriers to health care or health delivery to Deaf populations that exclude any health outcome data<br>• Papers reporting on health risk behaviours unless containing data on specific health outcomes |
| Country | No restriction by country | |
| Language | Research articles not published in English language nor a signed language | |
| Timeframe | Up until 13th June 2023 | |

* Signing Deaf populations are defined using a sociological/cultural-linguistic definition, referring to a particular group, not defined by the audiological condition of not hearing, but rather those who use a signed language as their first or preferred language–and share a culture. There is not one global population but signing Deaf populations exist in each country.

Abbreviations: GAD-7, Generalised Anxiety Disorder assessment-7; PHQ-9, Patient Health Questinnaire-9; LGBTQ, Lesbian Gay Bisexual Transgender and Queer.

## Data extraction

Comprehensive data extraction was performed using a pre-specified data extraction tool based on the Cochrane data collection form (Collecting data—form for RCTs and non-RCTs) and additional data extraction criteria to accommodate a range of study designs. This included

extracting information on study samples, methodology, limitations, evidence gaps, results, and a quality assessment for critical appraisal. Three reviewers ([AUTHORS ONE, TWO, and THREE]) extracted data independently. Results were then compared and discussed, with any disagreements settled between them and an additional reviewer ([AUTHOR FIVE]).

## Quality assessment

The CASP Cohort Study checklist (https://casp-uk.net/casp-tools-checklists/) was used to assess bias in each study as it was more appropriate to the range of items than other more design-specific checklists in the CASP suite (examples of bias include: was the cohort recruited in an acceptable way?; was the outcome accurately measured to minimise bias?; and how precise are the results?). Two reviewers ([AUTHORS ONE AND TWO]) independently assessed the risk of bias, with quality checks performed on 25% of the extracted papers by a third reviewer ([AUTHOR FIVE]), to ensure consistency. Results were compared with any disagreements resolved by a third reviewer.

## Results

The number of records after duplicates were removed was 3,056. Following the screening of titles and abstracts, 2,984 were excluded, leaving 72 full-text articles to be assessed for eligibility. Of those, 37 were excluded. Reasons for exclusion were: wrong outcome (n = 8), wrong population (n = 22), wrong language (n = 1), wrong publication type (n = 5), and records not retrieved (n = 1). A total of 35 have been included in the review (Fig 1).

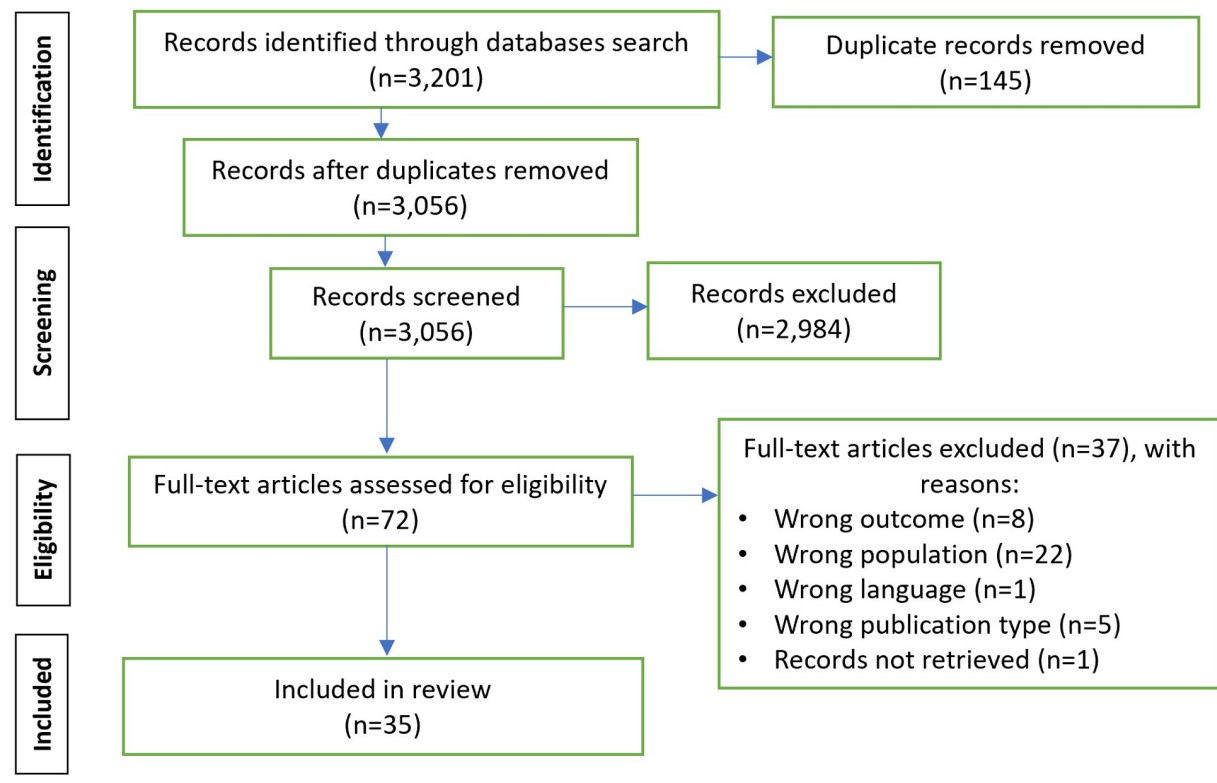

**Fig 1. PRISMA flow diagram: The findings from the searches.**

## Study characteristics

Of the 35 included studies, 15 were carried out in the USA, 6 in the UK, 3 in Sweden, 2 in Norway, and 1 each in Australia, Brazil, New Zealand, Napal, France, Thailand, Spain, Denmark, and Austria. Full details of study characteristics of the 35 included articles are shown in Table 2. All 35 items included were peer review journal articles of which 27 reported primary data and 8 were secondary data analyses. 29 studies utilised self-report health data. The identified publications varied by research setting and sample size. 32 out of the 35 studies took place in economically well resourced, developed countries, 2 from low- and middle-income countries and 1 from least developed countries. Only one study involved a randomised controlled trial.

Of the 35 studies, 26 used comparator populations in their study designs. Of these, 19 used comparisons with the "general population". This was not always distinguished to mean the hearing population. For example, the general population could include people who had a hearing loss but were not signing Deaf people. Comparators were either general population reference data for a particular disease, or general population survey study samples (e.g. Health Survey of England data). Three studies [15–17] used external datasets from the general population to construct a comparison group that were to some degree matched for a range of demographic variables (e.g. age and gender). Eight studies sought comparative data between d/Deaf populations. For example, Deaf sign language users versus other deaf people who used spoken language or within Deaf communities whereby Deaf people were distinguished by intersecting characteristics such as ethnicity or sexuality.

## Study appraisal

Quality appraisal by study is shown in Table 3. Application of the two initial screening questions in CASP resulted in 28 of 35 studies being eligible for quality appraisal with tool. No studies met all the CASP criteria. Comments on the strengths/weaknesses of studies are incorporated in the presentation of health outcome data below.

## Health conditions studied

The summary of each study's findings are reported in Table 4. The coverage of health conditions represented in the included articles is described according to the International Classification of Diseases 11[th] Revision (ICD-11) with some studies encompassing more than one of the 26 main categories. 15 out of the 26 classifications are encompassed within the 35 included studies. The greatest number of studies (25 out of 35) concern ICD-11 Code 6 Mental, Behavioural or Neurodevelopmental disorders. This includes anxiety/depression, mental wellbeing, psychiatric disorder, mental distress, and schizophrenia (See Table 5 for reported health outcomes by ICD-11 main classification code.). The remaining 11 classifications where there are no outcome studies identified are: 01 Certain infectious or parasitic; 03 Diseases of the blood or blood-forming organs; 04 Diseases of the immune system; 08 Diseases of the nervous system; 09 Diseases of the visual system; 10 Diseases of the ear or mastoid process; 14 Diseases of the skin; 17 Conditions related to sexual health; 19 Certain conditions originating in the perinatal period; 25 Codes for special purposes; and 26 Supplementary Chapter Traditional Medicine Conditions—Module I.

## Health outcomes in comparison with hearing/general populations

Comparison with hearing/general populations were used in 19 studies. The findings of these comparison are presented in the categories by health outcomes. An overview of health outcomes in these studies is reported in Table 6.

**Table 2. Study characteristics.**

| Author, year, and country | ICD-11 code | Health outcome | Aim | Sample | Inclusion criteria | Exclusion criteria |
|---|---|---|---|---|---|---|
| Ammons et al (2020); USA | 24 | Quality of life | To report quantitative results from CDC's BRFSS Caregiving Module survey from deaf informal caregivers of loved ones with Alzheimer's Disease and Related Dementias (ADRD) | 44 participants, 66 years, 79% female | Not reported | Not reported |
| Anderson et al (2021); USA | 06 | Perinatal depression | To create an initial ASL translation of the Edinburgh Postnatal Depression Scale (EPDS) for use among Deaf perinatal women | 36 participants, 31.6 years, 100% female | Deaf ASL user, currently pregnant or within one year postpartum, aged ≥18 years | Lack of capacity to consent |
| Barnett et al (2011); USA | 05; 23 | Obesity; suicide | To develop and administer an ASL-accessible health survey to estimate deaf individuals' health status and health risk compared to the local general population as a means of identifying health inequities | 339 participants, 46.4 years, 54.5% female | Not reported | Not reported |
| Barnett et al (2016); USA | 23 | Suicide | To compare the Rochester Deaf Health Survey 2013 findings with the Behavioural Risk Factor Surveillance System completed among the general adult population | 211 participants, 44.7 years, 37.4% female | Not reported | Not reported |
| Barnett et al (2023); USA | 05; 06; 11 | Obesity; cholesterol; diabetes; depression /anxiety; hypertension; cardiovascular disease | To address the absence of evidence-based weight-control programs developed for use with Deaf people | 104 participants, 53.5 years, 71% female | Deaf ASL users, BMI of 25 to 45 from community settings in Rochester, New York | Individuals experiencing pregnancy, breastfeeding, or planning a pregnancy. |
| Belk et al (2016); UK | 06 | Depression /anxiety | To determine the appropriate clinical cut-offs of the PHQ-9 BSL and GAD-7 BSL when used with Deaf people who sign and examine their operating characteristics | Dataset 1: 502 participants, 42 years, 60.4% female Dataset 2: 85 participants, 40 years, 57.6% female | Deaf sign language user, ≥16 years, accessed BSL-IAPT service since December 2011, received step 2 or 3 service, attended ≥1 therapist contact session | Not reported |
| Chapman et al (2017); Denmark | 06 | Mental wellbeing | To examine how different forms of identity are associated with levels of psychological well-being | 246 participants, 39.9 years, 51.2% female | Not reported | Not reported |
| Crowe et al (2016); Nepal | 06 | Psychiatric disturbance | To report the findings of the mental health needs and community support systems for deaf and hard of hearing adults in Nepal | 99 participants, 27.5 years, 36% female | Not reported | Not reported |
| Druel et al (2018); France | 02 | Cancer | To assess the average diagnostic stage of cancer in the Deaf community and discuss deafness as a contributing factor | 80 participants, 54.5 years, 42.5% female | Deaf community member, diagnosed with cancer between 01/01/2005 and 31/12/2014 | Non-Deaf patients, patients with pre-cancerous lesions or with cancer recurrence, origin of cancer not identified, aged <18 |
| Duarte et al (2021); Brazil | 24 | Quality of Life | To evaluate the psychometric properties of WHOQOL-Brief in the Brazilian Sign Language version (WHOQOL-Bref/ Libras) | 311 participants, 36.6 years, 55% female | Aged 18–65 years, communication by Libras | Not reported |

*(Continued)*

**Table 2.** (Continued)

| Author, year, and country | ICD-11 code | Health outcome | Aim | Sample | Inclusion criteria | Exclusion criteria |
|---|---|---|---|---|---|---|
| Ehn et al (2018); Sweden | 06; 20; 23 | Psychological health problems; Usher syndrome type 1; suicide | To explore the relation between work, health, social trust, and financial situation in Usher Syndrome 1 | 47 participants, 44 years, 53% female | Registered on the Swedish Usher database | Persons >65, reporting ≥1 disability |
| Emond et al (2014); UK | 05; 06; 11; 12 | Obesity; cholesterol; diabetes; depression; hypertension; cardiovascular disease; respiratory conditions | To assess the current health of the Deaf community in the UK compared with the general population | 298 participants, age NR, 53% female | Not reported | Not reported |
| Fellinger et al (2005); Austria | 06; 24 | Psychiatric disorder; Quality of Life | To assess mental distress and quality of life in a large community sample of deaf people in Upper Austria using adapted standardised instruments | 233 participants, 45.3 years, 44% female | Not reported | Not reported |
| Henning et al (2011); New Zealand | 24 | Quality of Life | To explore usage and accessibility of sign language interpreters, appraise the levels of quality of life of deaf adults residing in New Zealand, and consider the impact of sign language interpreters on QOL | 68 participants, 42.2 years, 63% female | Not reported | Not reported |
| Horton (2010); USA | 06 | Schizophrenia | To examine the role of linguistic ability in relation to cognition, social cognition, and functional outcome among deaf adults with schizophrenia or schizoaffective disorder | 34 participants, 45 years, 38% female | Not reported | Not reported |
| James et al (2022); USA | 05; 06 | Obesity; depression; | To examine differences in health risk behaviours, concerns, and access to health care among Deaf ASL users in Florida | 92 participants, 43.2 years, 65% female | Self-identifying as Deaf, hard of hearing, Deaf-blind, Deaf-plus or hearing impaired, using ASL to communicate, aged ≥18 years, residing in Florida | Not reported |
| James et al. (2022); USA | 06; 12; 18; 21; 22; 23 | Depression /anxiety; respiratory conditions; pregnancy; abdominal pain; superficial injury; suicide | To examine inequalities with emergency department utilization outcomes among DHH patients compared to non-DHH English speaking patients | 277 participants, 48.3 years, 58.4% female | DHH ASL users | Not reported |
| Kushalnagar et al (2019); USA | 02; 06;12; 15 | Cancer; depression /anxiety; respiratory conditions; arthritis; | To explore whether within-group disparity exists for medical conditions among Deaf LGBTQ older adults compared to Deaf non-LGBTQ counterparts | Non-LGBTQ: 803 participants, 63 years, 62.5% female LGBTQ: 178 participants, 58 years, 58.4% female | ASL as primary language, aged ≥ 18 years | Aged ≤18 years, unilateral hearing loss |
| Kushalnagar et al (2019); USA | 06; 21; 24 | Depression /anxiety; chronic comorbidity; health status | To examine the prevalence of self-reported depression and anxiety disorder diagnosis in a US sample of Deaf adults | 1704 participants, 48 years, 59% female | ASL as primary language, aged ≥ 18 female | Not reported |

(*Continued*)

**Table 2.** (Continued)

| Author, year, and country | ICD-11 code | Health outcome | Aim | Sample | Inclusion criteria | Exclusion criteria |
|---|---|---|---|---|---|---|
| Kushalnagar et al (2020); USA | 02; 05; 06; 11; 12 | Cancer; diabetes; depression /anxiety; hypertension; cardiovascular disease; respiratory conditions; | To explore adverse childhood communication experiences and their relative risks for acquiring specific chronic diseases and mental health disorders later in life | 1524 participants, 46 years, 59% female | Deaf or hard of hearing adults who were born deaf or became deaf before aged 13 years | Not reported |
| Kvam et al (2007); Norway | 06 | Depression /anxiety | To disclose the mental health situation among Deaf individuals compared to hearing individuals | 431 participants, 50.2 years, 59% female | Not reported | Not reported |
| McKee et al (2014); USA | 11 | Cardiovascular disease | To examine whether educational attainment and/or household income is inversely associated with cardiovascular risk among Deaf ASL users | 302 participants, age NR, 55.3% female | Not reported | Not reported |
| Munro et al (2009); Australia | 06 | Mental wellbeing | To examine the reliability, validity and acceptability of an Auslan version of the Outcome Rating Scale (ORS-Auslan) | Non-clinical: 55 participants, 49 years (median), 60.5% female Clinical: 44 participants, 39.5 years (median), 60% female | Deaf Auslan user, ≥18 years | Not reported |
| Øhre et al (2017); Norway | 06 | Mental distress | To compare the distribution of mental illness symptoms, disorders, and demographic characteristics among Deaf and hard of hearing patients using Norwegian Sign Language (NSL) and DHH patients using spoken Norwegian | 40 participants, 35.6 years, 62.5% female | Deaf and hard of hearing patients using Norwegian Sign Language (NSL), aged ≥18 years | Aged <18 years, dual sensory loss requiring tactile communication, referral for reasons other than assessment and treatment of mental disorders, acute and severe psychiatric or somatic illness |
| Peñacoba et al (2020); Spain | 06 | Depression /anxiety; mental wellbeing | To explore the possible relationship between emotional intelligence and psychological well-being in a sample of Deaf Spanish adults | 146 participants, 43 years, 58% female | Spanish citizens, aged ≥18 years, proficient in SSL and/or SOL, diagnosis of mild to profound hearing loss | Clinical diagnosis of any mental disorder |
| Perrodin-Njoku et al (2021); USA | 02; 05; 06; 11; 12; 15 | Cancer; obesity; diabetes; depression /anxiety; hypertension; respiratory conditions; arthritis | To understand the prevalence of health outcomes in a Black DHH adult sample compared to a Black hearing sample | 204 participants, age NR, 63% female | Aged ≥18 years old, Black, DHH community members who use ASL as primary language | Aged <18 years old, unilateral hearing loss |
| Rogers et al (2013); UK | 06 | Depression /anxiety | To translate 3 widely used clinical assessment measures (the Patient Health Questionnaire (PHQ-9), the Generalized Anxiety Disorder 7-item (GAD-7) scale, and the Work and Social Adjustment Scale (WSAS) into British Sign Language (BSL), pilot the BSL versions, and establish their validity and reliability | 136 participants, age NR, 60% female | Deaf BSL users, aged ≥16 years, residing in the United Kingdom | Not Deaf, not BSL users, learning disability, psychosis, current inpatients on mental health wards, unable to access signed information through the visual interface (i.e., Deafblind) |

(*Continued*)

**Table 2.** (Continued)

| Author, year, and country | ICD-11 code | Health outcome | Aim | Sample | Inclusion criteria | Exclusion criteria |
|---|---|---|---|---|---|---|
| Rogers et al (2016); UK | 24 | Health status | To translate the English version of EQ-5D-5L into BSL; validate the EQ-5D-5L BSL on a Deaf population; investigate the psychometric properties and establish its reliability | 92 participants, age NR, 69.6% female | Deaf BSL users, aged ≥18 years, residing in the United Kingdom | Not reported |
| Rogers et al (2018); UK | 24 | Health status | To translate the original English version of the Short Warwick-Edinburgh Mental Wellbeing Scale into BSL, examine its psychometric properties, and establish its validity and reliability | 96 participants, 42 years, 69% female | Self-reported Deaf BSL users, aged ≥18 years, residing in the United Kingdom | Unable to use an online interface to complete the questionnaire (e.g., severely visually impaired). |
| Sanfacon et al (2021); USA | 02; 05; 06; 11; 12; 15 | Cancer; diabetes; depression /anxiety; hypertension; cardiovascular disease; respiratory conditions; arthritis | To identify characteristics associated with medical conditions, including depression and anxiety disorders, among Deaf transgender adults | 74 participants, 37 years, 77% female | Deaf adults who were born deaf or became deaf before aged 13 years | Not reported |
| Shields et al (2020); UK | 06; 24 | Global distress; health status | To assess whether responses to the EQ-5D differ between Deaf BSL users and the general population, whether socio-demographic characteristics and clinical measures are associated with EQ-5D index scores, and the impact of psychological distress and depression on health status | 92 participants | Deaf BSL users, aged ≥18 years, residing in the United Kingdom | Not reported |
| Simons et al (2018); USA | 11 | hypertension | To determine the prevalence of hypertension in deaf American Sign Language (ASL) users | 532 participants, age NR, 53% female | Deaf ASL users | Not reported |
| Vichayanrat et al (2014); Thailand | 13 | Oral health | To determine dental caries status, oral hygiene, and oral health related behaviours among Deaf college students from Ratchasuda College, Thailand | 97 participants, 22.1 years, | Aged ≥18 years, attending Rachasuda College studying for BA (major in deaf studies) and Diploma programmes (major in sign language interpreter) | Not reported |
| Wahlqvist et al (2016); Sweden | 06; 20; 21; 23 | Mental health; Usher syndrome type 1; physical health; suicide | To describe the physical and psychological health, as well as social trust and financial situation of persons with USH1 | 60 participants, 49 years, 60% female | USH1, USH 1B and 1D clinical diagnosis | Not reported |

(*Continued*)

**Table 2.** (Continued)

| Author, year, and country | ICD-11 code | Health outcome | Aim | Sample | Inclusion criteria | Exclusion criteria |
|---|---|---|---|---|---|---|
| Werngren-Elgström et al (2003); Sweden | 05; 06; 07; 12; 15; 16 | Metabolism; depression; insomnia; respiratory conditions; musculoskeletal problems; Gastrointestinal-urinary tract symptoms | To investigate health-related quality of life, and the prevalence of depressive symptoms and insomnia among elderly pre-lingually deaf persons who use sign language | 45 participants, 75 years, 58% female | Aged ≥65 years, pre-lingually deaf, sign language users, living in Scania. | Not reported |

ADRD—Alzheimer's and related dementias, ASL—American sign language, BRFSS—The Behavioural Risk Factor Surveillance System, BSI—Brief Symptom Inventory, CCI—Charlson Comorbidity Index, CDC—Centre for Disease Control and Prevention, CORE-OM—Clinical Outcomes in Routine Evaluation–Outcome Measure, DASS-21—Depression Anxiety Stress Scale-21, DHH—Deaf and hard of hearing, DHS—Deaf Health Survey, ELCEs—Deaf Profile Early Life Communication Experiences, EPDS—Edinburgh Postnatal Depression Scale, EQ-5D-5L—European Quality of Life 5 Dimensions 5 Level Version, GAD-7—The Generalised Anxiety Disorder Assessment, GAF—Global Assessment of Functioning Scale, GDS—Geriatric Depression Scale, GHQ-12—General Health Questionnaire, HADS—Hospital Anxiety and Depression Scale, HET—Health on Equal Terms, HINTS—Health Information National Trends Survey in ASL, MINI—Mini International Neuropsychiatric Interview, NHIS—National Health Interview Survey, NSL—Norwegian sign language, PHQ-9—Patient Health Questionnaire -9, PROMIS—Patient Reported Outcomes Measurement Information System, PWBS—Psychological Well-Being Scale, RAMH—Rapid Assessment of Mental Health Needs, SCL-25 Hopkins Symptom Checklist, SRQ-20—Self-Reporting Questionnaire, SWEMWBS—Short Warwick-Edinburgh Mental Well-Being Scale, TAS-20—Toronto Alexithymia Scale, TMMS-24—Trait Meta-Mood Scale, USH-1—Usher Syndrome 1, WHO-5—World health Organisation Well-Being Index, WHOQOL-Bref—The World Health Organization Quality of Life Instrument, Short Form

**Cancer.** A large-scale study [18] involving secondary data from medical records of Deaf cancer patients found that Deaf people were diagnosed at a more advanced stage of colorectal and prostate cancer (64% of the Deaf group vs 13% of the reference group for prostate cancer, and 100% of the Deaf people were diagnosed at stage III/IV vs 47% for the reference group for colorectal cancer). The Deaf group had larger tumours at the time of diagnosis and their cancers were more likely to have spread to lymph nodes or metastasised to other organs. Deaf people were also more likely to be diagnosed with larger tumours in breast cancers (T2+ size was 60% for Deaf people compared to the reference group 34%) which is related to poorer prognosis, although there was no difference in the metastatic spread between the groups [18]. In an age-matched large-scale study, Perrodin-Njoku et al. [17] found that Black Deaf people were more likely to have cancer overall compared to Black hearing people (OR = 3.53, CI 1.61–7.71).

**Obesity.** Two primary studies [8, 9] recruiting 339 and 298 respectively reported significantly higher rates of overweight/obesity among Deaf populations when compared to published data on general populations: 35% of Deaf adults vs 26.6% of adults in the US general population [9] and 72% of Deaf men and 71% of Deaf women were overweight or obese vs 65% of men and 58% of women in the general population in the UK [8] HSE dataset. In the Emond et al. study [8], 90% of the over 65 Deaf group were overweight or obese. Neither a small-scale patient record study (n = 92) [19] nor a case-matched comparator study in Black populations [17] reported any significant differences. However, differences in the mean age of the comparator groups was noted for both studies which could help to explain the non-significant findings.

**Cholesterol.** Emond et al. [8] found that the mean level for both males and females was 'considerably' lower compared to general population Health Survey of England (HSE) reference data [8] although it was not clear whether the reported difference is statistically significant or not. No potential co-variates were examined.

**Table 3. CASP study appraisal for each included in the review.**

| Author(s) (year) | CASP criterion | | | | | | | | | | | | |
|---|---|---|---|---|---|---|---|---|---|---|---|---|---|
| | 1 | 2 | 3 | 4 | 5(a) | 5(b) | 6(a) | 6(b) | 8 (precise) | 9 | 10 | 11 | 12 |
| **Ammons et al (2020)** [37] | Y | CT | Y | Y | Y | Y | Y | Y | N | CT | CT | Y | Y |
| **Anderson et al (2021)** [33] | Y | Y | Y | Y | CT | Y | Y | Y | N | CT | CT | CT | Y |
| **Barnett et al (2011)** [9] | Y | Y | Y | Y | Y | CT | CT | CT | Y | Y | N | Y | Y |
| **Barnett et al (2016)** [28] | Y | Y | Y | Y | Y | N | CT | Y | Y | CT | N | CT | Y |
| **Barnett et al (2023)** [44] | Y | Y | Y | Y | Y | Y | Y | Y | Y | Y | CT | Y | Y |
| **Belk et al (2016)** [43] | Y | Y | Y | Y | Y | N | Y | Y | N | CT | Y | CT | Y |
| **Chapman et al (2017)** [35] | Y | Y | Y | CT | Y | Y | Y | Y | N | Y | Y | CT | Y |
| **Crowe et al (2016)** [62] | Y | CT | CT | CT | CT | N | CT | Y | N | CT | N | CT | CT |
| **Druel et al (2018)** [18] | Y | Y | Y | Y | CT | Y | Y | Y | N | Y | Y | Y | Y |
| **Duarte et al (2021)** [38] | Y | Y | Y | Y | Y | Y | CT | Y | Y | Y | Y | Y | Y |
| **Ehn et al (2018)** [15] | Y | Y | Y | Y | CT | CT | Y | Y | Y | Y | Y | Y | Y |
| **Emond et al (2015)** [8] | Y | Y | Y | Y | CT | CT | Y | Y | N | Y | Y | Y | N |
| **Fellinger et al (2005)** [30] | Y | Y | N | CT | Y | CT | Y | Y | N | N | CT | Y | Y |
| **Henning et al (2011)** [31] | Y | CT | CT | Y | Y | Y | Y | Y | N | Y | CT | Y | Y |
| **Horton (2010)** [63] | Y | CT | Y | Y | Y | Y | Y | Y | N | Y | CT | Y | Y |
| **James et al (2022)** [19] | Y | Y | Y | Y | Y | CT | Y | Y | Y | CT | CT | Y | Y |
| **James et al (2022)** [25] | Y | Y | Y | CT | Y | Y | Y | Y | Y | CT | Y | Y | Y |
| **Kushalnagar et al (2019)** [32] | Y | Y | CT | CT | Y | Y | Y | Y | Y | Y | Y | CT | Y |
| **Kushalnagar et al (2019)** [20] | Y | Y | Y | Y | Y | Y | Y | Y | Y | Y | Y | CT | Y |
| **Kushalnagar et al (2020)** [39] | Y | Y | Y | Y | Y | Y | Y | Y | Y | Y | Y | CT | Y |
| **Kvam et al (2007)** [21] | Y | Y | Y | CT | Y | Y | Y | Y | Y | CT | Y | Y | CT |
| **McKee et al (2014)** [36] | Y | Y | Y | CT | Y | Y | Y | Y | Y | Y | CT | Y | Y |
| **Munro et al (2009)** [41] | Y | CT | Y | CT | N | N | Y | Y | Y | Y | CT | Y | Y |
| **Øhre et al (2017)** [40] | Y | Y | Y | CT | Y | N | CT | CT | N | CT | CT | Y | Y |
| **Peñacoba et al (2020)** [16] | Y | Y | Y | N | Y | Y | Y | Y | Y | Y | Y | Y | Y |
| **Perrodin-Njoku et al (2022)** [17] | Y | Y | Y | CT | Y | CT | Y | Y | Y | Y | Y | CT | Y |
| **Rogers et al (2013)** [42] | Y | CT | Y | Y | Y | N | Y | Y | N | Y | CT | CT | Y |
| **Rogers et al (2016)** [29] | Y | Y | Y | Y | Y | N | Y | Y | Y | Y | N | Y | Y |
| **Rogers et al (2018)** [23] | Y | Y | Y | Y | Y | N | Y | Y | Y | Y | N | CT | Y |
| **Sanfacon et al (2021)** [34] | Y | Y | Y | Y | Y | Y | Y | Y | Y | Y | Y | CT | Y |
| **Shields et al (2020)** [10] | Y | Y | Y | Y | Y | Y | Y | Y | Y | Y | CT | Y | Y |
| **Simons et al (2018)** [24] | Y | Y | Y | Y | Y | Y | Y | Y | Y | Y | Y | N | Y |
| **Vichayanrat et al (2014)** [26] | Y | CT | Y | CT | Y | Y | Y | Y | N | CT | CT | N | Y |
| **Wahlqvist et al (2016)** [27] | Y | Y | Y | CT | Y | Y | Y | Y | Y | CT | Y | Y | CT |
| **Werngren-Elgström et al (2003)** [22] | Y | Y | Y | Y | CT | N | Y | Y | N | Y | CT | CT | Y |

Note: Y = yes, CT = Can't tell, N = no. CASP criterion 7 reports the results and is excluded here as they are included in the main text. In the CASP criterion 8 (precise) Y indicates Confidence Intervals (CIs) were reported, and N that they were not.

**Diabetes.** Edmond et al. [8] found that of those Deaf people who reported diabetes, at least half of the participants' diabetes was not under control, which could lead to higher rates of diabetic complications. Perrodin-Njoku et al. [17] reported that Black Deaf people are more likely to have diabetes compared to Black hearing people in the US (OR = 1.77, CI = 1.04–3.02). The type of diabetes was not reported in either study.

**Depression/Anxiety.** The prevalence of depression / anxiety in Deaf adults was found to be significantly higher compared with the hearing population. A large-scale study (n = 1,704)

**Table 4. Study outcomes.**

| Author, year, and country | ICD-11 code | Health outcome | Primary outcome and measurement | Key outcomes |
|---|---|---|---|---|
| Ammons et al (2020); USA | 24 | Quality of life | Quality of Life (BRFSS Caregiver Module, PROMIS-Deaf Profile measures) | • More years of education are significantly associated with higher generic quality of life (p < .04) and deaf-specific quality of life (p < .02)<br>• More years of caregiving are significantly associated with worse deaf-specific quality of life (p < .03) |
| Anderson et al (2021); USA | 06 | Perinatal depression | Depression severity (Edinburgh Postnatal Depression Scale—EPDS) | • Participants reported a mean total score of 5.6 out of 30 points on the ASL-EPDS (SD = 4.2)—31% scored in the mild, 6% moderate, and 0% severe depression ranges<br>• No measured sociodemographic characteristics were significantly associated with EPDS score, but racial/ethnic status approached significance (p = 0.142) |
| Barnett et al (2011); USA | 05; 23 | Obesity; suicide | Health status and health risk (Behavioural Risk Factor Surveillance System (BRFSS) in ASL) | • Obesity prevalence was higher among the Deaf sample compared to the general population (34.2% versus 26.6%), but significance was not reported<br>• Past year suicide attempt prevalence was higher among the Deaf sample compared to the general population (2.2% versus 0.4%), but significance was not reported |
| Barnett et al (2016); USA | 23 | Suicide | Health risk (Rochester Deaf Health Survey; BRFSS) | • Obesity prevalence was lower among the Deaf sample compared to the general population (29.5% versus 29.6%), but significance was not reported<br>• Past year suicide attempt prevalence was higher among the Deaf sample compared to the general population (1.5% versus 0.5%) but significance was not reported |
| Barnett et al (2023); USA | 05; 06; 11 | Obesity; cholesterol; diabetes; depression/anxiety; hypertension; cardiovascular disease | Changes in weight and depression symptoms (biometric measures, PHQ-9) | • At 6 months, the difference in mean weight change for the immediate-intervention arm versus the delayed-intervention arm (no intervention yet) was-3.4 kg (= 0.0424)<br>• Most (61.6%) in the immediate arm lost ≥5% of baseline weight versus 18.1% in the no-intervention-yet arm (p< 0.001)<br>• 39.6% (n = 40) of the sample had PHQ-9 scores indicative of at least mild depression (<4)<br>• Self-reported prevalence was reported for diabetes (11.5%), high cholesterol (53.8%), heart attack (1%), and coronary heart disease (2.9%) |
| Belk et al (2016); UK | 06 | Depression/anxiety | Health status (PHQ-9 BSL, GAD-7 BSL) | • PHQ-9 BSL scores—dataset 1 (IAPT) = 14.58 (SD 5.99) versus dataset 2 (previous study of PHQ-9 and GAD-7) = 3.62 (SD 3.29)<br>• GAD-7 BSL scores—dataset 1 (IAPT) = 12.50 (SD = 4.98) versus dataset 2 = 2.13 (SD 2.48) |

*(Continued)*

**Table 4.** (Continued)

| Author, year, and country | ICD-11 code | Health outcome | Primary outcome and measurement | Key outcomes |
|---|---|---|---|---|
| Chapman et al (2017); Denmark | 06 | Mental wellbeing | Psychological wellbeing (World health Organisation Well-Being Index (WHO-5)) | • Deaf (65.5), hearing (66), and bicultural identity (66.9) was associated with significantly higher levels of psychological wellbeing, compared to marginal identity (46.9) (P = <0.05)<br>• Those with marginal identity (52.9%) were significantly more likely to report additional disability than those with Deaf identity (27.2%) (P = <0.05) |
| Crowe et al (2016); Nepal | 06 | Psychiatric disturbance | Mental health need (Self-Reporting Questionnaire (SRQ-20), Rapid Assessment of Mental Health Needs (RAMH)) | • Mean SRQ-20 score was 6.70, which is below the cut-off score for mental health problems. However, 38% met the threshold for mental health problems<br>• 24% of the sample met the threshold for the presence of negative environmental influences |
| Druel et al (2018); France | 02 | Cancer | Cancer stage (multidisciplinary team diagnosis, pathologist summary) | • Breast cancer diagnoses did not differ significantly between the Deaf sample and general population (P = 0.58)<br>• Advanced diagnosis of prostate (P = 0.04) and colorectal cancers (P = 0.03) were significantly higher among the Deaf sample compared to the general population |
| Duarte et al (2021); Brazil | 24 | Quality of Life | Quality of life (The World Health Quality of Life Instrument brief instrument (WHOQOL-Bref/Libras)) | • Scores across domains included: Physical health (0.641), Psychological (0.705), Environment (0.710), and Overall-Bref domains (0.873)<br>• Years of schooling were significantly associated with the psychological (P = <0.05), and physical, social, and environment domains (p≤ 0.001); sex was significantly associated with physical and psychological domains (p < 0.05); monthly income was significantly associated with the physical domain (p≤ 0.001) |
| Ehn et al (2018); Sweden | 06; 20; 23 | Psychological health problems; Usher syndrome type 1; suicide | Physical health, mental health, and social trust (Swedish Health on Equal Terms questionnaire) | • Compared to the general population reference group, individuals in the USH1 work group had an increased odds of psychological health (OR 3.86) or physical health (OR 6.93) problems, suicidal thoughts (OR 3.60), suicide attempts (OR 15.23), but none were significantly different<br>• Compared to the general population reference group, individuals in the USH1 non-work group had an increased odds of psychological health (OR 3.86) or physical health (OR 6.93) problems, suicidal thoughts (OR 3.60), suicide attempts (OR 15.23), but none were significantly different |

(*Continued*)

**Table 4.** (*Continued*)

| Author, year, and country | ICD-11 code | Health outcome | Primary outcome and measurement | Key outcomes |
|---|---|---|---|---|
| Emond et al (2014); UK | 05; 06; 11; 12 | Obesity; cholesterol; diabetes; depression; hypertension; cardiovascular disease; respiratory conditions | Physical health (standard health check and questionnaire) | • Participants self-reported prevalence rates for depression (24%), COPD (<1%), asthma (15% male, 17% female), and diabetes (7%)<br>• Rates of overweight/obese were significantly (p<0.001) higher among the Deaf compared to the general population (72% versus 65% for males, 71% versus 58% females)<br>• Elevated blood pressure was significantly (p<0.001) higher among the Deaf compared with the general population (37% versus 21%)<br>• Self-reported rates of cardiovascular disease were significantly less among the Deaf population compared to the general population (11.1% versus 26.2% for those aged 65–82 years) (P<0.01) |
| Fellinger et al (2005); Austria | 06; 24 | Psychiatric disorder; Quality of Life | Quality of Life (WHOQOL-BREF (The World Health Organisation's Brief Quality of Life questionnaire), General Health Questionnaire (GHQ-12), Brief Symptom Inventory (BSI)) | • The deaf sample has significantly poorer quality of life than the general population for the physical (68.13 versus 76.92) and psychological 64.16 versus 74.12) domains (p<0.01) measured by the WHOQOL-BREF<br>• All findings with the GHQ-12 (4.38 versus 1.16) and the BSI show significantly higher levels (p = 0.01) of emotional distress among the deaf. |
| Henning et al (2011); New Zealand | 24 | Quality of Life | Quality of life (WHOQOL-BREF) | • The Deaf sample scored significantly lower than a comparable hearing sample on all four WHOQOL-BREF domains (p<0.01), including physical (26.20 versus 27.27), psychological (22.00 versus 23.7), social relationships (10.98 versus 11.93) and environment (28.12 versus 33.70)<br>• Ease of access for interpreters was a significant predictor for physical (2.56, p<0.02) and environment (-4.05, p<0.00) domains |
| Horton (2010); USA | 06 | Schizophrenia | Linguistic ability, schizophrenia prevalence | • Linguistic ability is positively and significantly associated with functional outcome (p <0.10)<br>• Younger age of sign language acquisition is significantly associated with superior linguistic ability, but did not moderate the effect of linguistic ability on other domains (p<0.006) |
| James et al (2022); USA | 05; 06 | Obesity; depression; | Health risk (self-report survey) | • Mental health was the most reported health concern among Deaf people (28.6%)<br>• A higher prevalence of mental health concern was observed among adults aged 18–29 than among adults aged ≥40 (33.3% vs 25.0%)<br>• Rates of overweight/obesity were higher among the Deaf sample compared to the hearing sample (69.5% versus 66.7%), but significance was not reported<br>• 15.5% of the Deaf sample scored ≥3 on the PHQ-2 (indicating depression) |

(*Continued*)

**Table 4.** (Continued)

| Author, year, and country | ICD-11 code | Health outcome | Primary outcome and measurement | Key outcomes |
|---|---|---|---|---|
| James et al. (2022); USA | 06; 12; 18; 21; 22; 23 | Depression/anxiety; respiratory conditions; pregnancy; abdominal pain; superficial injury; suicide | Emergency department utilisation and health status (frequency of visits, Charlson Comorbidity Index) | • Emergency department admissions were higher among Deaf-ASL participants than non-deaf English speakers for several variables; including abdominal pain (10.8% versus 6.2%) and pregnancy complications (0.7% versus 0.5%), but significance was not reported<br>• DHH English speakers, but not ASL users, had higher odds than non-DHH English speaking patients of using the ED in the past 12 months (OR—1.79)<br>• DHH ASL users (~9–10%) represented fewer ED encounters for injury-related ED visits compared to DHH English speakers (~52–59%) and non-DHH English speakers (~33–38%) |
| Kushalnagar et al (2019); USA | 02; 06;12; 15 | Cancer; depression/anxiety; respiratory conditions; arthritis; | Physical and mental health (Health Information National Trends Survey in ASL) | • Significant differences between Deaf LGBTQ and non-LGBTQ groups emerged for several medical conditions, including chronic lung diseases($p = 0.01$), depression/anxiety ($p = 0.001$), and cancer history ($p = 0.05$)<br>• Self-identification as LGBTQ was significantly associated with increased risks for chronic lung diseases (RR = 1.74), arthritis (RR = 1.26), depression/anxiety (RR = 1.71), and comorbidity (RR = 1.25) |
| Kushalnagar et al (2019); USA | 06; 21; 24 | Depression/anxiety; chronic comorbidity; health status | Mental health prevalence (Health Information National Trends Survey ASL) | • Rates of depression/anxiety were significantly ($p<0.01$) higher for Deaf adults compared to hearing adults (24.9% versus 21.7%)<br>• The hearing sample had more individuals with comorbidity and worse overall health status compared to the Deaf sample ($p<0.001$)<br>• Significant ($p<0.001$) differences in health status were observed between the Deaf and hearing sample for excellent /good (43.3% versus 58.3%), good (40.2% versus 33.9%) and fair/poor (16.5% versus 7.8%) |
| Kushalnagar et al (2020); USA | 02; 05; 06; 11; 12 | Cancer; diabetes; depression/anxiety; hypertension; cardiovascular disease; respiratory conditions; | Physical and mental health (Patient-Reported Outcomes Measurement Information System (PROMIS), Deaf Profile Early Life Communication Experiences (ELCEs)) | • Lifetime prevalence for medical conditions was: 32% for diabetes, 8% for heart conditions, 32% for hypertension, 16% for lung condition, 27% for depression/anxiety disorders, and 10% for cancer<br>• 55% of the sample perceived their health status as being very good/excellent, 34% good, 11% poor/ fair<br>• Poorer direct child–caregiver communication was significantly ($p = <0.05$) associated with diabetes (RR = 1.12) hypertension (RR = 1.10), and heart disease (RR = 1.61)<br>• Poor indirect family communication/ inclusion increased risks for lung diseases (RR = 1.19) and depression/anxiety (RR = 1.34) |

*(Continued)*

**Table 4.** (Continued)

| Author, year, and country | ICD-11 code | Health outcome | Primary outcome and measurement | Key outcomes |
|---|---|---|---|---|
| Kvam et al (2007); Norway | 06 | Depression/anxiety | Mental health (Hopkins Symptom Checklist (SCL-25)) | • Differences between the distribution of answers (not at all, a little, quite a bit, extremely) were significantly ($p = <0.001$ different between the hearing and Deaf sample for feeling fearful, hopeless, and feeling blue.<br>• Women were significantly more anxious than men across both groups ($p = <0.001$)<br>• Younger respondents reported significantly ($p = <0.001$) more feelings of hopelessness and feeling blue<br>• The odds of an individual experiencing mental distress are significantly ($p = <0.001$) greater among those who are Deaf |
| McKee et al (2014); USA | 11 | Cardiovascular disease | Cardiovascular risk/event prevalence (Deaf Health Survey (DHS), Behavioural Risk Factor Surveillance System (BRFSS)) | • Deaf respondents who reported high school education or less were more likely to report the presence of a cardiovascular disease equivalent (OR 5.76) compared to Deaf respondents who reported having a >4 year college degree, significance not reported<br>• Deaf respondents who reported annual income of <$25,000 were not significantly more likely to report the presence of a cardiovascular disease equivalent (OR 2.24) compared to Deaf respondents who reported annual incomes of >$50,000 |
| Munro et al (2009); Australia | 06 | Mental wellbeing | Reliability and acceptability of translated measures (ORS-Auslan visual analogue measure, Depression Anxiety Stress Scale-21 (DASS-21)) | • The clinical sample had a significantly lower mean ($p<0.001$), indicating lower levels of well-being in the clinical sample (18.57 versus 27.04).<br>• Using the DASS-21-Auslan, the non-clinical sample had mean scores of 15.10 total, 4.69 for depression, 3.55 for anxiety and 7.18 for stress subscales |
| Øhre et al (2017); Norway | 06 | Mental distress | Mental health (Mini International Neuropsychiatric Interview (MINI), Symptom Check List-25 (SCL-25), Global Assessment of Functioning Scale (GAF)) | • The frequency of psychiatric comorbidity was similar in the two linguistic groups (40% versus 36%) but there were no statistically significant differences between groups regarding age at onset of mental disorder<br>• Significantly more Norwegian speaking than signing patients reported comorbid medical conditions ($p<0.01$)<br>• In both groups, SCL mean scores on anxiety and depression were above recommended threshold for mental disorder. |
| Peñacoba et al (2020); Spain | 06 | Depression/anxiety; mental wellbeing | Emotional intelligence and psychological wellbeing (Trait Meta-Mood Scale (TMMS-24), Alexithymia: Toronto Alexithymia Scale (TAS-20), Anxiety and Depression: Hospital Anxiety and Depression Scale (HADS), Psychological Well-Being Scale (PWBS)) | • Significant ($p = <0.05$) differences were found between deaf and hearing participants regarding anxiety (8.06 versus 6.60), depression (5.01 versus 3.27), alexithymia (21.51 versus 14.37), and psychological well-being (24.58 versus 27.44)<br>• Significant ($p = <0.05$) differences found between educational level and overall psychological well-being<br>• Significant ($p = <0.05$) correlations between mental health conditions (anxiety and depression) and psychological well-being |

*(Continued)*

**Table 4.** (Continued)

| Author, year, and country | ICD-11 code | Health outcome | Primary outcome and measurement | Key outcomes |
|---|---|---|---|---|
| Perrodin-Njoku et al (2021); USA | 02; 05; 06; 11; 12; 15 | Cancer; obesity; diabetes; depression/anxiety; hypertension; respiratory conditions; arthritis | Physical health (National Cancer Institute's Health Information National Trends Survey (HINTS)) | • Deaf participants had significantly more lung disease than hearing group (19.9% versus 13.4%) (p< = 0.05)<br>• Hearing participants had significantly more heart conditions than the Deaf group (9.8% versus 8.3%) (p = <0.05)<br>• Black DHH adults had a higher likelihood for the following health conditions: diabetes (OR = 1.77), hypertension (OR = 2.72), lung disease (OR = 1.72), cancer (OR = 3.52,), and comorbidity (OR = 2.91)<br>• No group differences were observed for heart disease, arthritis, depression/anxiety, and obesity |
| Rogers et al (2013); UK | 06 | Depression//anxiety | Acceptability and reliability of translated measures (PHQ-9, GAD-7, WSAS, Clinical Outcomes in Routine Evaluation–Outcome Measure (CORE-OM)) | • For groups 1 and 2 combined, the mean scores on the items were: PHQ 9 (5.34), GAD-7 (3.25), WSAS (4.71), CORE-OM (0.76)<br>• Scores for group 2 (mental health difficulty in previous 12 months) were significantly (p = <0.01) higher than group 2 for the PHQ-9 (11.61 versus 3.25), GAD-7 (7.0 versus 2.0) and WSAS (12.93 versus 2.68), and CORE-OM (1.46 versus 0.53) |
| Rogers et al (2016); UK | 24 | Health status | Acceptability and reliability of translated measures (EuroQol (EQ-5D-5L)) | • Mean EQ-5D-5L BSL utility index median value was 0.78 and median score was 0.84<br>• The mean score for CORE-10 BSL is 11.74<br>• The percentage of the study sample with 'no problems' in each domain was less than UK population published norms: mobility (69% versus 82%); self-care (86% versus 96%); usual activities (61% versus 84%); discomfort (48% versus 67%), and anxiety (46% versus 79%) |
| Rogers et al (2018); UK | 24 | Health status | Acceptability and reliability of translated measures (Short Warwick-Edinburgh Mental Well-Being Scale (SWEMWBS)) | • Mean SWEMWBS BSL baseline score was 22.82<br>• CORE-OM BSL well-being subscale had a mean score of 1.35<br>• EQ-5D VAS BSL had a mean score of 68.0<br>• Those currently experiencing mental health difficulties had a significantly (p = <0.001) lower (worse) mean score compared to those not currently experiencing mental health difficulties |
| Sanfacon et al (2021); USA | 02; 05; 06; 11; 12; 15 | Cancer; diabetes; depression/anxiety; hypertension; cardiovascular disease; respiratory conditions; arthritis | Physical and mental health (translated tool created by the researchers) | • Lifetime prevalence was reported at 48.6% for depression/anxiety disorders, 28.8% for hypertension, 20.3% for lung conditions,16.2% for arthritis, 12.3% for diabetes, 7.0% for cirrhosis/liver/kidney problems, 5.5% for heart conditions, and 2.7% for cancer<br>• Identification as nonbinary increased risk of depression/anxiety by 80% relative to binary gender<br>• Lifetime prevalence was significantly different between binary and non-binary groups only for depression and anxiety disorder (37.5% versus 63.6%) (p = <0.05) |

*(Continued)*

**Table 4.** (Continued)

| Author, year, and country | ICD-11 code | Health outcome | Primary outcome and measurement | Key outcomes |
|---|---|---|---|---|
| Shields et al (2020); UK | 06; 24 | Global distress; health status | Acceptability and reliability of translated measures (EQ-5D-5L, CORE10/CORE-6D) | • 73% of Deaf participants reported some problems on one or more EQ-5D dimensions<br>• For Deaf with complete data, the proportions reporting no problems were 70% mobility, 88% self-care, 63% usual activities, 49% pain/discomfort and 47% anxiety/depression<br>• EQ-VAS score was lower than the mean general population (70.23 versus 79.15)<br>• The overall mean EQ-5D-5L index values were lower than the general population (0.78 versus 0.84)<br>• 58% of participants had psychological distress and 43% met the criteria for depression |
| Simons et al (2018); USA | 11 | hypertension | Hypertension prevalence (HINTS, Cycle 4) | • The age-weighted prevalence for hypertension was significantly lower in the deaf sample compared with the hearing sample (33% versus 46%) (p = 0.001)<br>• •Significant age-weighted group differences were found for age (p = 0.001), race (p = 0.04), and education (p = <0.01)<br>• The Deaf sample demonstrated a significantly (p = <0.01) decreased risk for hypertension compared with the hearing sample (37% versus 45%) |
| Vichayanrat et al (2014); Thailand | 13 | Oral health | Dental caries prevalence (oral examination) | • No significant difference was observed between the hearing and deaf for caries prevalence, DMFT, and oral hygiene status |
| Wahlqvist et al (2016); Sweden | 06; 20; 21; 23 | Mental health; Usher syndrome type 1; physical health; suicide | General health (Health on Equal Terms (HET) questionnaire) | • The psychological health, social trust, and financial situation of persons with USH1 were significantly (p = <0.05) poorer compared to the general population<br>• Persons with USH1 reported significantly (p = <0.05) poorer psychological health across all 8 variables, including fatigue (62% versus 49%), loss of confidence (16% versus 6%), suicidal thoughts (30% versus 12%), suicide attempts (16% versus 4%), unhappiness (19% versus 11%), not facing problems (18% versus 9%)<br>• No significant differences were reported for physical health, except for hand, elbow, leg and knee pain which was significantly higher among the general population (43% versus 23%) (p = 0.003) |

(*Continued*)

**Table 4.** (Continued)

| Author, year, and country | ICD-11 code | Health outcome | Primary outcome and measurement | Key outcomes |
|---|---|---|---|---|
| Werngren-Elgström et al (2003); Sweden | 05; 06; 07; 12; 15; 16 | Metabolism; depression; insomnia; respiratory conditions; musculoskeletal problems; Gastrointestinal-urinary tract symptoms | Depressive symptom and insomnia prevalence (15-item version of the geriatric depression scale (GDS), Livingston's sleep scale) | • One third of the deaf persons demonstrated depressive symptoms and nearly two thirds suffered from insomnia. There was substantial correlation between insomnia, depressive symptoms and lower subjective wellbeing.<br>• Depressive symptoms were significantly (p = <0.01) higher among the Deaf sample compared with the general population (51% versus 29%)<br>• Feelings of restlessness were significantly higher (p = <0.01) among the Deaf sample compared with the general population (44% versus 18%) |

ADRD—Alzheimer's and related dementias, ASL—American sign language, BRFSS—The Behavioural Risk Factor Surveillance System, BSI—Brief Symptom Inventory, CCI—Charlson Comorbidity Index, CDC—Centre for Disease Control and Prevention, CORE-OM—Clinical Outcomes in Routine Evaluation–Outcome Measure, DASS-21—Depression Anxiety Stress Scale-21, DHH—Deaf and hard of hearing, DHS—Deaf Health Survey, ELCEs—Deaf Profile Early Life Communication Experiences, EPDS—Edinburgh Postnatal Depression Scale, EQ-5D-5L—European Quality of Life 5 Dimensions 5 Level Version, GAD-7—The Generalised Anxiety Disorder Assessment, GAF—Global Assessment of Functioning Scale, GDS—Geriatric Depression Scale, GHQ-12—General Health Questionnaire, HADS—Hospital Anxiety and Depression Scale, HET—Health on Equal Terms, HINTS—Health Information National Trends Survey in ASL, MINI—Mini International Neuropsychiatric Interview, NHIS—National Health Interview Survey, NSL—Norwegian sign language, PHQ-9—Patient Health Questionnaire -9, PROMIS—Patient Reported Outcomes Measurement Information System, PWBS—Psychological Well-Being Scale, RAMH—Rapid Assessment of Mental Health Needs, SCL-25 Hopkins Symptom Checklist, SRQ-20—Self-Reporting Questionnaire, SWEMWBS—Short Warwick-Edinburgh Mental Well-Being Scale, TAS-20—Toronto Alexithymia Scale, TMMS-24—Trait Meta-Mood Scale, USH-1—Usher Syndrome 1, WHO-5—World health Organisation Well-Being Index, WHOQOL-Bref—The World Health Organization Quality of Life Instrument, Short Form

by Kushalnagar et al. [20] reported 24.9% compared with 21.7% and that it occurred at an earlier age; Kvam et al.' Norwegian population study [21] reported 33.8% compared with 6.8%; Peñacoba et al. [16] reported mean scores for anxiety of 8.06 compared with 6.60 and for depression 5.01 compared with 3.27 in a case-matched study of Spanish Deaf and hearing adults. Peroddin-Njoku et al. [17] reported no difference in the Black Deaf population compared to the Black hearing population, they stated that the issue of medical mistrust in the general Black community might be a factor in finding no difference between the two groups. Werngren-Elgström et al. [22] in a small Swedish comparative study (n = 45) found that Deaf people aged 65 and over have a higher prevalence of depression compared to their hearing counterparts (37% vs 23%).

**Mental well-being.** A case-matched study of Spanish Deaf adults (n = 146) reported significantly lower psychological well-being compared to Spanish hearing people: mean score of 24.58 vs 27.44 [16]. Rogers et al. [23] small scale validation study reported a non -significant lower well-being mean score on the SWEMWBS (22.82) in comparison with the general population (23.64).

**Hypertension.** The frequency of raised blood pressure was significantly higher for Deaf people (37%) compared to the HSE data (21%) although the confidence interval was not reported [8]. Perrodin-Njoku et al. [17] also reported that Black Deaf people are more likely to experience higher blood pressure compared to the Black hearing population (OR = 1.73). However, a large-scale study (n = 532) by Simons et al. [24] reported that the prevalence for

**Table 5. Reported health outcomes in the included studies by ICD-11 main classification code.**

| ICD-11 Code | Health outcome | Studies |
|---|---|---|
| 02 Neoplasms | Cancer | Druel et al (2018) [18]<br>Kushalnagar et al (2019) [32]<br>Kushalnagar et al (2020) [39]<br>Perrodin-Njoku et al (2022) [17]<br>Sanfacon et al (2021) [34] |
| 05 Endocrine, nutritional, or metabolic diseases | Obesity | Barnett et al (2011) [9]<br>Barnett et al (2023) [44]<br>Emond et al (2015) [8]<br>James et al (2022) [19]<br>Perrodin-Njoku et al (2022) [17] |
| | Cholesterol | Barnett et al (2023) [44]<br>Emond et al (2015) [8] |
| | Diabetes | Barnett et al (2023) [44]<br>Emond et al (2015) [8]<br>Kushalnagar et al (2020) [39]<br>Perrodin-Njoku et al (2022) [17]<br>Sanfacon et al (2021) [34] |
| | Metabolism symptoms—feeling cold/overweight/etc. | Werngren-Elgström et al (2003) [22] |
| 06 Mental, behavioural, or neurodevelopmental disorders | Perinatal depression | Anderson et al (2021) [33] |
| | Depression / Anxiety | Belk et al (2016) [43]<br>Barnett et al (2023) [44]<br>James et al (2022) [25]<br>Kushalnagar et al (2019) [32]<br>Kushalnagar et al (2019) [20]<br>Kushalnagar et al (2020) [39]<br>Kvam et al (2007) [21]<br>Peñacoba et al (2020) [16]<br>Perrodin-Njoku et al (2022) [17]<br>Rogers et al (2013) [42]<br>Sanfacon et al (2021) [34] |
| | Depression | Emond et al (2015) [8]<br>James et al (2022) [19]<br>Werngren-Elgström et al (2003) [22] |
| | Mental well-being | Chapman et al (2017) [35]<br>Munro et al (2009) [41]<br>Peñacoba et al (2020) [16]<br>Rogers et al (2018) [23] |
| | Psychiatric disturbance / RAMH—info inc. Mental health needs | Crowe et al (2016) [62] |
| | HET questionnaire–Psychological health problems | Ehn et al (2018) [15] |
| | Psychiatric disorder / Psychopathology | Fellinger et al (2005) [30] |
| | Schizophrenia | Horton (2010) [63] |
| | Mental distress / Distress / Functioning | Øhre et al (2017) [40] |
| | Global distress | Shields et al (2020) [10] |
| | Mental health (including fatigue / loss of confidence / constant tension / worthlessness / not facing up to problems / unhappiness) | Wahlqvist et al (2016) [27] |
| 07 Sleep-wake disorders | Insomnia | Werngren-Elgström et al (2003) [22] |
| 11 Diseases of the circulatory system | Hypertension / Blood pressure | Barnett et al (2023) [44]<br>Emond et al (2015) [8]<br>Kushalnagar et al (2020) [39]<br>Perrodin-Njoku et al (2022) [17]<br>Sanfacon et al (2021) [34]<br>Simons et al (2018) [24] |
| | Cardiovascular disease | Barnett et al (2023) [44]<br>Emond et al (2015) [8]<br>Kushalnagar et al (2020) [39]<br>McKee et al (2014) [36]<br>Sanfacon et al (2021) [34] |

*(Continued)*

**Table 5.** (Continued)

| ICD-11 Code | Health outcome | Studies |
|---|---|---|
| 12 Diseases of the respiratory system | Respiratory conditions | **Emond et al (2015)** [8] <br> **James et al (2022)** [25] <br> **Kushalnagar et al (2019)** [32] <br> **Kushalnagar et al (2020)** [39] <br> **Perrodin-Njoku et al (2022)** [17] <br> **Sanfacon et al (2021)** [34] <br> **Werngren-Elgström et al (2003)** [22] |
| 13 Diseases of the digestive system | Oral health | **Vichayanrat et al (2014)** [26] |
| 15 Diseases of the musculoskeletal system or connective tissue | Arthritis | **Kushalnagar et al (2019)** [32] <br> **Perrodin-Njoku et al (2022)** [17] <br> **Sanfacon et al (2021)** [34] |
| | Musculo-skeletal symptoms | **Werngren-Elgström et al (2003)** [22] |
| | Spondylosis / intervertebral disc disorders / other back problems | **James et al (2022)** [25] |
| 16 Diseases of the genitourinary system | Gastrointestinal-urinary tract symptoms | **Werngren-Elgström et al (2003)** [22] |
| 18 Pregnancy, childbirth, or the puerperium | Other complications of pregnancy / haemorrhage during pregnancy / abruptio placenta / placenta previa / other complications of birth / spontaneous abortion | **James et al (2022)** [25] |
| 20 Developmental anomalies | Usher syndrome type 1 | **Ehn et al (2018)** [15] <br> **Wahlqvist et al (2016)** |
| 21 Symptoms, signs, or clinical findings, not elsewhere classified | Abdominal pain / Nonspecific chest pain | **James et al (2022)** [25] |
| | Chronic comorbidity | **Kushalnagar et al (2019)** [32] <br> **Kushalnagar et al (2019)** [20] |
| | Physical health: headache / tinnitus / hand, elbow, knee and leg pain | **Wahlqvist et al (2016)** [27] |
| 22 Injury, poisoning, or certain other consequences of external causes | Superficial injury / contusion / sprains and strains / open wounds of extremities / other injuries and conditions due to external causes / open wounds of the head, neck, and trunk | **James et al (2022)** [25] |
| 23 External causes of morbidity or mortality | Suicide attempts / suicidal thoughts | **Barnett et al (2011)** [9] <br> **Barnett et al (2016)** [28] <br> **Ehn et al (2018)** [15] <br> **James et al (2022)** [25] <br> **Wahlqvist et al (2016)** [27] |
| 24 Factors influencing health status or contact with health services | Health status | **Kushalnagar et al (2019)** [20] <br> **Rogers et al (2016)** [29] <br> **Rogers et al (2018)** [23] <br> **Shields et al (2020)** [10] |
| | Quality of life | **Ammons et al (2016)** [37] <br> **Duarte et al (2021)** [38] <br> **Fellinger et al (2005)** [30] <br> **Henning et al (2011)** [31] |

hypertension was significantly lower in the Deaf sample (33%) compared with 46% in the hearing sample.

**Cardiovascular disease.** Cardiovascular disease was significantly less self-reported by Deaf people compared to the general population [8]. Emond et al. [8] found that treatment rate for Deaf men of all CVD was 45% compared with treatment rate for ischaemic heart disease and stroke of between 61% and 70% for men age aged 55–84 in the general population.

**Respiratory / lung conditions.** Self-reported Chronic Obstructive Pulmonary Disease (COPD) was less in Deaf adults (1%) compared to the HSE data (4% of men and 5% of women) [8] although the significant difference was not stated. Black Deaf people have a greater likelihood of developing a lung condition compared with Black hearing people (OR = 1.72) [17]. Fewer DHH ASL users were reported in emergency department encounters for lower respiratory disease compared to DHH English speakers and hearing English speakers (n = 11 vs n = 62 and n = 29 respectively) [25].

**Table 6. Overview of health outcomes in comparison with hearing / general populations.**

| Health outcome | Signing Deaf populations in comparisons with hearing/general population samples |
|---|---|
| Cancer | • Diagnosed at a more advanced stage (Druel et al., 2018) [18].<br>• Higher risk of cancer overall for Black Deaf people when compared with the general Black population (Perrodin-Njoku et al., 2022) [17]. |
| Obesity | • Increased prevalence (Barnett et al., 2011; Emond et al., 2015) [9, 8].<br>• No difference in weight/obesity prevalence (James et al., 2022; Perrodin-Njoku et al., 2022) [17, 19]. |
| Cholesterol | • 'Considerably' lower than the general population reference data (Emond et al., 2015) [8]. |
| Diabetes | • Similar prevalence to the general population (Emond et al., 2015) [8] but more likely to be uncontrolled.<br>• Higher prevalence amongst Black Deaf people when compared to Black hearing people (Perrodin-Njoku et al., 2022) [17]. |
| Depression / Anxiety | • Prevalence of anxiety/depression is higher in Deaf adults (Kushalnagar et al., 2019; Kvam et al., 2007; Peñacoba et al., 2020) [16, 20, 21] and depression in older Deaf adults (Wengren-Elgström et al., 2003) [22].<br>• No difference amongst Black Deaf people and Black hearing people (Perrodin-Njoku et al., 2022) [17]. |
| Mental well-being | • Deaf adults scored significantly lower (Peñacoba et al., 2020; Rogers et al., 2018) [16, 23]. |
| Hypertension | • Higher blood pressure (Emond et al., 2015; Perrodin-Njoku et al., 2022) [8, 17].<br>• Lower prevalence of high blood pressure (Simons et al., 2018) [24]. |
| Cardiovascular disease | • Lower prevalence reported (Emond et al., 2015) [8]. |
| Respiratory conditions | • Lower prevalence reported of COPD (Emond et al., 2015) [8]. |
| Lung condition | • Black Deaf people have greater likelihood of developing a lung condition when compared with Black hearing people (Perrodin-Njoku et al., 2022) [17]. |
| Oral health | • No difference in oral hygiene status, prevalence of cavities or DMHFT (Decayed, Missing and Filled Teeth) (Vichayanrat et al., 2014) [26]. |
| Arthritis | • No significant difference in prevalence amongst Black Deaf people when compared with Black hearing people (Perrodin-Njoku et al., 2022) [17]. |
| Chronic comorbidity | • Fewer co-morbidities reported in Deaf people (Kushalnagar et al., 2019) [20]. |
| Headache | • More prevalence for Deaf people with USH1 (Wahlqvist et al., 2016) [27]. |
| Suicide attempts / suicidal thoughts | • Higher prevalence of attempts (Barnett et al., 2011; Barnett et al., 2016) [9, 28].<br>• Deaf people with Usher Syndrome Type 1 were more likely to attempt suicide (Ehn et al., 2018; Wahlqvist et al., 2016) [15, 27]. |
| Health status | • Poorer health status amongst Deaf people (Kushalnagar et al., 2019; Rogers et al., 2016; and Shields et al., 2020) [10, 20, 29]. |
| Quality of Life | • Lower quality of life (Fellinger et al., 2005; Henning et al., 2011) [30, 31]. |
| Musculo-skeletal symptom | • Fewer reports of emergency department encounters for spondylosis, intervertebral disc disorders, and other back problems (James et al., 2022) [25]. |
| Pregnancy, childbirth or the puerperium | • Fewer reports of emergency department encounters for complications during pregnancy (James et al., 2022) [25]. |

**Oral health.** Vichayanrat et al. [26] reported no differences between Deaf and hearing people in prevalence of caries or DMFT (Decayed, Missing or Filled Teeth), and similar oral hygiene status. Those Deaf people who took part in Vichayanrat et al. [26] study were educated at BA and/or Diploma level, therefore, unlikely to be representative of the Deaf population.

**Arthritis.** A self-report study by Perrodin-Njoku et al. [17] found no significant difference in prevalence of arthritis between Black Deaf people and Black hearing people.

**Musculo-skeletal symptom.** James et al. [25] found that reporting of emergency department encounters for spondylosis, intervertebral disc disorders, and other back problems, was less for DHH ASL users (n = 19) in comparison with DHH English speakers (n = 56) and hearing English speakers.

**Pregnancy, childbirth or the puerperium.** Of the 32 encounters recorded in emergency department records for other complications of pregnancy James et al. [25], only 3 were DHH ASL users compared with 25 DHH English and 4 hearing English speakers.

**Headache.** A secondary data study by Wahlqvist et al. [27] reported that Deaf people with USH1 (n = 60) expressed significantly more prevalent problems with headaches compared to the cross section of the Swedish population including those with and without visual difficulties (n = 5738) (40% vs 26% respectively).

**Chronic comorbidity.** Kushalnagar et al. [20] reported that the hearing sample has more individuals with comorbidities compared to the Deaf sample (40.5% vs 34.2%), although the hearing sample was older than the Deaf sample which could explain the higher prevalence in the hearing sample.

**Suicide attempts / suicidal thoughts.** The prevalence of suicide attempts in the past year is higher in the Deaf population (2.2%) in the US than observed in the general population (0.4%) [9] and Deaf people reported more suicide attempts in the past year compared with the general population (1.5% vs 0.5%) [28]. Deaf people with Usher Syndrome Type 1 (USH1) are more likely to attempt suicide compared with the general population (16% vs 4%) [27]. James et al., [25] in a study of emergency department records report no suicide ideation and intentional self-inflicted injury reported for DHH ASL users, in comparison with two were reported for DHH English speakers and three for hearing English speakers [25].

**Health status.** Using validated assessments in sign language, health status was found to be poorer in the Deaf population compared with the general population in the self-report studies by Rogers et al. [29] (EQ-5D mean index values 0.78 vs 0.84), and Shields et al. [10] (43% vs 17% for depression symptoms). However, Kushalnagar et al. [20], found that hearing people had worse overall health status compared with Deaf people, suggesting that age may be a contributing factor, as the mean age of the hearing sample was significantly older than that of the Deaf adults. The Wahlqvist et al. [27] study reported that the USH1 group have greater problems with fatigue (62% versus 49%), and a loss of confidence (16% versus 6%) compared to the general population.

**Quality of life.** Fellinger et al. [30] and Henning et al. [31] both reported significantly lower Quality of Life as measured by WHOQOL-BREF in Deaf people compared with general populations. The use of the sign language version of WHOQOL-BREF was not reported in Fellinger et al. [30] study.

## Factors identified as influencing health outcomes within the Deaf population

**LGBTQ+ status.** Kushalnagar et al. [32] found that the Deaf LGBTQ population in the US, in comparison with the Deaf non-LGBTQ population, are more likely to have a personal cancer history (24.1% vs 15.2%), more likely to have a lung condition (23.4% vs 15%), and significantly more likely to experience depression/anxiety (33.3% vs 17.9%). Deaf LGBTQ status was also significantly associated with increased risk for arthritis (RR = 1.26) and for chronic comorbidity (2 or more medical conditions) (RR = 1.25) [32] in comparison with the Deaf non-LGBTQ population. A small-scale study (n = 36) reported that the LGBTQ

status were not significantly related to the depression score [33]. In a study involving transgender Deaf communities, it was found depression/anxiety was higher for those with nonbinary identities [34].

**Educational level.** Deaf people with university level education scored higher on psychological well-being compared with other Deaf people [16]. In another study, educational levels were found to be significant in explaining psychological well-being score [35]. A secondary data study found that Deaf people who reported low educational levels were more likely to be at risk for cardiovascular disease compared with Deaf people with a four-year college degree or more (OR = 5.76) [36]. Two small-scale self-report studies [37, 38] found that more years of education was significantly associated with higher quality of life for Deaf people.

**Employment and economic status.** Small-scale self-report study found that Deaf people who are not in employment have significantly lower mental well-being compared to those who are in employment (SWEMWBS BSL mean score 21.10 vs 23.40) [23]. Wahlqvist et al. [27] report those with USH1 who are in employment are more likely to attempt suicide compared to the general population who are in employment but those with USH1 who are not in employment the differences in suicidal thoughts are not significant compared to the non-working group in the general population [15]. Income status was reported not to have the presence of cardiovascular disease [36].

**Ethnicity.** Although some studies include race/ethnicity when describing the study samples, few studies have considered the influence of ethnicity on health outcomes. Perrodin-Njoku et al. [17] identified consistently poor health outcomes for Black Deaf adults with regard to diabetes, hypertension, heart condition, lung disease, and cancer, as well as comorbidity. Anderson et al. [31] reported that individuals who identified as a racial/ethnic minority significantly had slightly higher levels of perinatal depression than those who identified as White non-Hispanic. Kushalnagar et al. [20] reported no significant difference in race/ethnicity on depression/anxiety outcomes.

**Gender/sex.** Health outcomes by gender/sex were explored in a few studies. Significantly poorer physical well-being outcomes were reported for Deaf females in the validated sign language version study (n = 311) [38]. Poorer well-being / quality of life outcomes for Deaf females compared to Deaf males are found [8, 16, 21, 23, 30]. Kushalnagar et al. [20] higher prevalence of depression/anxiety amongst Deaf females. Deaf men were found to have significantly higher blood pressure (15.9%) compared to Deaf women (7.7%) [8].

**Language and communication.** Using inadequate access to direct child-caregiver communication in childhood as the independent variable, a large-scale study by Kushalnagar et al. [39] identified that it increased a person's risks of having diabetes by 12%, hypertension by 10%, lung disease by 19% and cardiovascular disease by 61% and increased risk for depression/anxiety by 34% compared to those Deaf people who had adequate access to indirect family communication and inclusion [39]. No significant difference in the scores for mood or neurosis were found between those Deaf people who used sign language and deaf people who used spoken language [40].

**Family history/personal medical history.** Using the sign language version of the assessment and when the validation has been examined, Munro et al. [41] reported that a clinical sample had a significantly lower mean score for wellbeing (18.57; SD = 9.6) compared with a non-clinical sample (27.04, SD = 8.68) on the ORS-Auslan. Overall health status was found to be poorer for Deaf people with depression compared to those with no psychological distress or depression [10]. Rogers et al. [42] and Belk et al. [43] found that severity of depression and anxiety was worse for those who self-reported as having mental health difficulties compared to those who did not.

**Age.** It was reported that diagnosis of depression/anxiety was likely to be young in the large-scale study [20], however the age was reported not to have impact on depression outcomes in Deaf populations in the study by Duarte et al. [38].

## Discussion

The findings from this systematic review demonstrate that, in general, physical health and mental health outcomes in Deaf signing populations are worse when compared with general population samples. Additionally, the impact of a health condition on other health outcomes can created further health inequalities. For example, although not a comparison to the general population, Barnett et al. [44] study which involved a whole sample who were overweight/obese (BMI of 25 or greater) and the biometric outcomes were recorded by a research nurse, 13.5% had diabetes, 37.5% had high blood pressure, 53.8% had high cholesterol, 2.9% had heart disease, 39.6% had a PHQ-9 score indicative of at least mild depression. However, the strength and quality of the evidence available overall is questionable. Firstly, sample definition is poor with inconsistencies in reporting which add to the difficulties in collating and appraising data concerning health outcomes for Deaf adults. The main issues include inaccurate or imprecise descriptions of participants meaning it is hard to discern in some studies who are Deaf sign language users and some study populations incorporated children and young people without any clear distinction from adults in data subsets. Secondly, some studies do not report whether the health outcomes measured used validated standard instruments in sign language nor report potential issues associated with interpreter-mediated assessment and engagement, particularly with regards to self-reported health data. Thirdly, secondary data analysis comparisons with 'general population' data will include some participants who are deaf but not sign language users unless matched 'hearing' samples have been constructed. Fourthly, creating binary comparisons between Deaf sign language users and hearing/non-signing people can cover up issues of diversity and intersectionality within Deaf communities. Where comparison groups are matched on a range of demographic variables, these may still hide different circumstances associated with variables e.g. social determinants that are more prevalent amongst Deaf populations e.g. under-employment or direct discrimination.

Furthermore, gaps remain in the knowledge of specific health outcomes as there is no reported health outcome data for the Deaf population in 11 out of the 26 (42.3%) of the ICD-11 disease classification categories, including, for example, diseases of the immune system, visual system and nervous system which indicates clear deficits in health outcome data for this population. The bias towards studies concerning mental health might be in part explained by the longstanding development of specialist mental health services for deaf people in some countries such as the UK and US garnering funding for evidence-based practices. The major neglect of data on physical health outcomes might be related to the considerable difficulties in recording and extracting *routine* health data that is specific enough to differentiate Deaf people from anyone who is categorised with a hearing disability in routine health data collection [45]. For example James et al. [25] in a study on emergency department encounters, highlighted the possibility that DHH ASL users were being mis-recorded as DHH English speakers. The invisibility of the Deaf population within clinical records is likely to contribute to a lack of focus on whether their outcomes are similar to those of the bigger population of adults with a hearing loss or disability but who are not members of a cultural-linguistic minority whose engagement with health services is fundamentally mediated by problems of linguistic access and cultural competence [46]. In addition, the overwhelming majority of the included studies concern Deaf people who reside in economically well-resourced countries. Yet, nearly 80% of people who experience deafness, whether sign language users or not, reside in low- and middle- income countries [1].

It is also notable that some study designs are not present in this field which would enhance knowledge. For example, there is a lack of qualitive, exploratory designs specific to health outcomes, there is very little longitudinal study data (whether retrospective or prospective) that might start to reveal patterns of health inequalities experienced by individuals over time. Although there is some large-scale secondary data analysis based on routinely collected national data, most disease-specific studies rely on small population samples the biases of which are not clearly examined. There is also a lack of information on inclusive research designs that involves Deaf populations at all stages of the research process.

The reasons for the health inequalities experienced by Deaf individuals are multiple and complex, both access to and communication with health services and clinicians are commonly cited problems [7]. Around 5% of deaf children have one or more parents who are d/Deaf, meaning that the vast majority are born to hearing parents, who usually have little experience of deafness and often have little or no knowledge of signed languages [47]. Age-appropriate literacy remains a key barrier to accessing information for a great many d/Deaf people and is especially apparent amongst sign language users of previous generations whose access to and quality of education has been particularly poor [48, 49]. Socio-economic inequalities are a common experience faced by Deaf people, for example, Deaf people are more likely to be unemployed and those who are in employment are likely to experience inequalities in earnings [50]. These issues of inequalities in language development, educational attainment, employment, and income would have impact across this group's lifetime relating to the likely disposable resources Deaf people have such as quality of life and opportunities. Prilleltensky et al. [51] has argued that without opportunities to access rights and equality, a person would be unable to fulfil their well-being potential, and that "social justice can help manage social determinants of health in a more equitable manner" [52, p.8]. Despite the legal rights to promote equality and combat discrimination in some countries (e.g. the Equality Act 2010 in the UK, and Americans with Disabilities Act in the USA) and the legal status of sign languages in some countries, Deaf people are likely to be faced with discrimination whether be it direct or indirect. The responsiveness of health services and health interventions to provide and promote understanding of health conditions in a signed language is also identified as inadequate in many countries. Deaf individuals are up to 7 times more likely to experience poor health literacy than their hearing counterparts, something which is closely tied to unhealthy behaviours, limited healthcare seeking behaviours, decreased service use and poorer health outcomes [53–55]. Studies show that Deaf people have limited knowledge of the symptoms of certain medical emergencies, such as heart attacks and strokes, and that in the US, only 61% would contact the emergency services in such cases [56]. Research has also explored the issue of inadequate adaptation of clinical and psychological assessment tools for use with Deaf patients [57, 58], resulting in both under and overdiagnosis of potentially serious health conditions and inadequate tracking of recovery [43, 59]. Understandably, Deaf populations have previously reported feelings of mistrust towards healthcare professionals [7], these populations are also found to be less likely to see the value in healthcare consultations when compared with the general population [60]. Aggravating this, many Deaf patients also have difficulty complaining about the healthcare barriers they face, as complaints processes often do not accommodate for sign language users [61]. Consequentially, healthcare professionals are unaware of the relevant issues, and no action is taken to amend them.

## Conclusion

This comprehensive systematic review on health outcomes in Deaf signing populations has highlighted health inequalities in comparison to general populations and within their own

communities. This review has highlighted the need for better classification in routine health records, better data on different kinds of health inequality, and better coverage of diseases to understand the Deaf experience and a wider range of study designs yielding different forms of evidence. Without understanding Deaf people's experience, it would be challenging to improve their healthcare and health outcomes. Addressing health inequalities in practice and research requires the inclusion of Deaf people's priorities for better health.

## Supporting information

**S1 Appendix. PROSPERO international prospective register of systematic reviews.** (PDF)

**S1 Checklist. PRISMA 2020 checklist.** (PDF)

**S1 Table. Keywords used in the search strategy.** (DOCX)

## Author Contributions

**Conceptualization:** Katherine D. Rogers, Gemma Shields, Alys Young.

**Data curation:** Katherine D. Rogers, Aleix Rowlandson.

**Formal analysis:** Katherine D. Rogers, Aleix Rowlandson, Gemma Shields, Alys Young.

**Funding acquisition:** Katherine D. Rogers.

**Investigation:** Katherine D. Rogers, Aleix Rowlandson, James Harkness, Alys Young.

**Methodology:** Katherine D. Rogers, Aleix Rowlandson, Gemma Shields, Alys Young.

**Project administration:** Katherine D. Rogers, Aleix Rowlandson.

**Resources:** Katherine D. Rogers, Aleix Rowlandson.

**Validation:** Katherine D. Rogers, Aleix Rowlandson, James Harkness, Gemma Shields, Alys Young.

**Visualization:** Katherine D. Rogers, Aleix Rowlandson.

**Writing – original draft:** Katherine D. Rogers, Aleix Rowlandson, James Harkness, Gemma Shields, Alys Young.

**Writing – review & editing:** Katherine D. Rogers, Aleix Rowlandson, James Harkness, Gemma Shields, Alys Young.

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
