## [Decision Letter · Decision Letter 0]

26 Feb 2024

PONE-D-24-02887Health outcomes in Deaf signing populations: a systematic reviewPLOS ONE

Dear Dr. Rogers,

Thank you for submitting your manuscript to PLOS ONE. After careful consideration, we feel that it has merit but does not fully meet PLOS ONE’s publication criteria as it currently stands. Therefore, we invite you to submit a revised version of the manuscript that addresses the points raised during the review process.

We look forward to receiving your revised manuscript.

Kind regards,

Ateya Megahed Ibrahim El-eglany

Academic Editor

PLOS ONE

Journal Requirements:

"This review is partly funded by Dr Katherine Rogers’s NIHR (National Institute of Health and Care Research) Post-Doctoral Fellowship (NIHR award reference number: PDF-2018-ST2-004). The views expressed in this publication are those of the author(s) and not necessarily those of the NIHR, NHS or the UK Department of Health and Social Care."

Reviewers' comments:

Reviewer's Responses to Questions

**Comments to the Author**

1. Is the manuscript technically sound, and do the data support the conclusions?

Reviewer #1: Yes

2. Has the statistical analysis been performed appropriately and rigorously? 

Reviewer #1: Yes

3. Have the authors made all data underlying the findings in their manuscript fully available?

Reviewer #1: Yes

4. Is the manuscript presented in an intelligible fashion and written in standard English?

Reviewer #1: Yes

5. Review Comments to the Author

Reviewer #1: Introduction

The introduction sets the stage but may lack a compelling narrative that highlights the nuances of health disparities faced by Deaf communities. It could benefit from integrating a broader range of literature to contextualize the unique health challenges and communication barriers Deaf people encounter. Additionally, specifying the research question or hypothesis at the outset could help sharpen the focus.

Methods

While following PRISMA guidelines lends credibility, the methodology section could improve by offering more clarity on the database search strategy, including search terms and inclusion/exclusion criteria. A detailed explanation of the study selection process (e.g., screening and eligibility assessment) and the rationale behind the chosen quality assessment tools for evaluating the included studies would enhance transparency and replicability.

Results

The results section is crucial for understanding the scope of health disparities among Deaf individuals. However, it could be more user-friendly. For instance, employing tables or figures to summarize the study characteristics and main findings could aid in visualizing the data. Discussing the heterogeneity of the study populations and the interventions assessed would also be valuable. A more nuanced examination of the results, considering different types of health outcomes (physical vs. mental health) and the quality of the evidence, is needed.

Discussion

The discussion adeptly identifies gaps in the research and potential implications. However, it could delve deeper into the mechanisms driving the observed disparities, such as access to healthcare, communication barriers, or socioeconomic factors. A comparison with other minority groups facing similar health disparities could offer additional insights. Moreover, a stronger emphasis on actionable recommendations for stakeholders at various levels (e.g., healthcare systems, policy, community engagement) would make this section more impactful.

Conclusion

The conclusion succinctly wraps up the review but could be more forceful in calling for action. Highlighting specific, actionable steps that different sectors can take to address the disparities identified could provide a clear path forward. Additionally, emphasizing the importance of inclusive research practices that involve Deaf communities in the creation of knowledge about their health could inspire more community-engaged research efforts.

Overall

General Observations for Improvement: Integrating more qualitative studies into the review could offer deeper insights into the lived experiences of Deaf individuals, providing a richer context for the quantitative findings. Additionally, addressing the potential for publication bias and discussing the implications of the findings in the context of current health policy debates would enhance the article's relevance and applicability.

A major revision would provide an opportunity to address these critiques by:

Clarifying and expanding on the methodology, particularly regarding study selection, quality assessment, and analysis techniques.

Improving the presentation and accessibility of results with visual aids and clearer categorization.

Deepening the discussion on the implications of findings, including a more thorough examination of social determinants and actionable recommendations for various stakeholders.

Strengthening the conclusion with concrete steps and highlighting the importance of inclusive research practices.

6. PLOS authors have the option to publish the peer review history of their article (what does this mean?). If published, this will include your full peer review and any attached files.

Reviewer #1: **Yes: **Mostafa shaban

---

## [Author Response · Author response to Decision Letter 0]

20 Mar 2024

Thank you to both academic editor and reviewer' comments. Please find the attached 'Response to reviewers'.

---

## [Decision Letter · Decision Letter 1]

22 Mar 2024

Health outcomes in Deaf signing populations: a systematic review

PONE-D-24-02887R1

Dear Dr. 

We’re pleased to inform you that your manuscript has been judged scientifically suitable for publication and will be formally accepted for publication once it meets all outstanding technical requirements.

Kind regards,

Ateya Megahed Ibrahim El-eglany

Academic Editor

PLOS ONE

Additional Editor Comments (optional):

Reviewers' comments:

Reviewer's Responses to Questions

**Comments to the Author**

1. If the authors have adequately addressed your comments raised in a previous round of review and you feel that this manuscript is now acceptable for publication, you may indicate that here to bypass the “Comments to the Author” section, enter your conflict of interest statement in the “Confidential to Editor” section, and submit your "Accept" recommendation.

Reviewer #1: All comments have been addressed

Reviewer #2: All comments have been addressed

2. Is the manuscript technically sound, and do the data support the conclusions?

Reviewer #1: Yes

Reviewer #2: Yes

3. Has the statistical analysis been performed appropriately and rigorously? 

Reviewer #1: Yes

Reviewer #2: Yes

4. Have the authors made all data underlying the findings in their manuscript fully available?

Reviewer #1: Yes

Reviewer #2: Yes

5. Is the manuscript presented in an intelligible fashion and written in standard English?

Reviewer #1: Yes

Reviewer #2: Yes

6. Review Comments to the Author

Reviewer #1: Dear author

thank you for your effort done to revise the paper

i recommend accepting the paper in the current status

Reviewer #2: (No Response)

7. PLOS authors have the option to publish the peer review history of their article (what does this mean?). If published, this will include your full peer review and any attached files.

Reviewer #1: **Yes: **Mostafa Shaban

Reviewer #2: **Yes: **Ateya Megahed Ibrahim

---

## [Editor Report · Acceptance letter]

1 Apr 2024

PONE-D-24-02887R1 

PLOS ONE

Dear Dr. Rogers, 

I'm pleased to inform you that your manuscript has been deemed suitable for publication in PLOS ONE. Congratulations! Your manuscript is now being handed over to our production team.

Kind regards, 

on behalf of

Dr. Ateya Megahed Ibrahim El-eglany 

Academic Editor

PLOS ONE